# The Effect of Vitamin E Supplementation on Serum Aminotransferases in Non-Alcoholic Fatty Liver Disease (NAFLD): A Systematic Review and Meta-Analysis

**DOI:** 10.3390/nu15173733

**Published:** 2023-08-25

**Authors:** Stamatina Vogli, Androniki Naska, Georgios Marinos, Maria-Iosifina Kasdagli, Philippos Orfanos

**Affiliations:** Department of Hygiene, Epidemiology and Medical Statistics, School of Medicine, National and Kapodistrian University of Athens, 11527 Athens, Greece; stamvog95@gmail.com (S.V.); gmarino@med.uoa.gr (G.M.); kasdaglimar@med.uoa.gr (M.-I.K.); phorfanos@med.uoa.gr (P.O.)

**Keywords:** non-alcoholic fatty liver disease, NAFLD, non-alcoholic steatohepatitis, NASH, aminotransferases, alanine transaminase, ALT, aspartate transaminase, AST, vitamin E, tocopherols, tocotrienols, systematic review

## Abstract

Νon-alcoholic fatty liver disease (NAFLD) is a common cause of end-stage liver disease in developed countries. Oxidative stress plays a key role during the course of the disease and vitamin E supplementation has shown to be beneficial due to its antioxidative properties. We aim to investigate the effect of vitamin E supplementation on serum aminotransferase levels in patients with NAFLD. Three electronic databases (MEDLINE, CENTRAL, and Embase) were reviewed for randomized trials that tested vitamin E supplementation versus placebo or no intervention in patients with NAFLD, published until April 2023. A total of 794 patients from 12 randomized trials were included in this meta-analysis. Notwithstanding the studies’ heterogeneity and moderate internal validity in certain cases, among studies testing vitamin E supplementation at 400 IU/day and above, the values of alanine aminotransferase (ALT) were reduced compared with placebo or no intervention [ALT Mean Difference (MD) = −6.99 IU/L, 95% CI (−9.63, −4.35), for studies conducted in Asian countries and MD = −9.57 IU/L, 95% CI (−12.20, −6.95) in non-Asian countries]. Regarding aspartate aminotransferase (AST), patients in the experimental group experienced a reduction in serum levels, though smaller in absolute values [AST MD = −4.65 IU/L, 95% CI (−7.44, −1.86) in studies conducted in Asian populations] and of lower precision in non-Asian studies [MD = −5.60 IU/L, 95% CI (−11.48, 0.28)].

## 1. Introduction

The prevalence of non-alcoholic fatty liver disease (NAFLD) is increasing throughout the world and it is estimated to reach 30% globally, with the highest rates being in Latin America (44%) and in the Middle East and North Africa (36.5%) [1]. In Europe, the pooled NAFLD prevalence in adults has been estimated to be 26.9% [2]. Few Asian studies have published NAFLD incidence rates and a meta-analysis of these studies conducted by Younossi et al. [3] estimated that the annual incidence rate for NAFLD in Asia was about 52 cases per 1000 person years. The NAFLD refers to a spectrum of liver disease ranging from the simple non-alcoholic fatty liver (NAFL), which is a deposition of adipose tissue in the liver defined by the presence of steatosis in >5% of hepatocytes, to the non-alcoholic steatohepatitis (NASH), which is a more progressive type of liver disease defined histologically by hepatocellular damage and is usually accompanied by hepatic fibrosis [4]. It is estimated that 11% of patients with NASH will progress to cirrhosis within 15 years and hepatocellular carcinoma (HCC) may develop in up to 13% of patients with NASH and cirrhosis [5,6]. In this context, NASH has been recognized as one of the leading causes of cirrhosis in adults and NASH-related cirrhosis is currently the second indication for liver transplantation in the United States [6,7].

NAFLD patients tend to be obese, with dyslipidemia, hypertriglyceridemia, hypertension, insulin resistance, and/or type 2 diabetes (T2D). The aforementioned features comprise the most important risk factors for NAFLD and are among the principal features of metabolic syndrome, making NAFLD the liver disease component of it [3]. Other independent risk factors for NAFLD include a diet high in carbohydrates and low in mono- and poly-unsaturated fatty acids and a single-nucleotide polymorphism of the PNPLA3 allele (commonly found in Hispanics), a gene that encodes a protein involved in lipid metabolism [3,8]. Regarding age, although NAFLD prevalence increases with age [3], the burden of NAFLD in children is also increasing following the alarming rising of obesity and metabolic syndrome in this age group [9,10].

Currently, a “multiple-hit hypothesis” that implicates numerous cellular and molecular mechanisms playing a role in a parallel or sequential manner is an accepted explanation of NAFLD pathogenesis [11,12]. Oxidative stress is increasingly recognized as a key factor contributing to NASH development [13] and levels of anti-oxidant enzymes and compounds, such as α-tocopherol, have been found to be significantly decreased in NASH patients [14,15] providing the rationale of the use of antioxidants in clinical trials to examine their potent hepatoprotective effects [15]. Vitamin E, the lipid-soluble anti-oxidant that includes both tocopherols and tocotrienols, has been studied in a number of clinical trials, administered at different doses and forms in NAFLD/NASH patients of all ages [16,17,18,19]. The rising evidence that vitamin E supplementation can result in improvement in steatosis and resolution of steatohepatitis in non-diabetic adults with NASH, led to the recently published guidelines of the American Association for the Study of Liver Diseases that recommend physicians to consider vitamin E administration at a dose of 800 IU/day in non-diabetic, non-cirrhotic adults with NASH [5].

In a systematic review and a meta-analysis of studies assessing the effect of vitamin E on liver dysfunction, including the most commonly used biochemical indices in NAFLD patients, i.e., serum aminotransferases such as alanine aminotransferase (ALT) and aspartate aminotransferase (AST), Vadarlis et al. [20] concluded that vitamin E supplementation can reduce the levels of aminotransferases compared with a placebo. Recently, new clinical trials have been published and we have therefore performed a systematic review and meta-analysis to study the effect of vitamin E supplementation as a single agent versus placebo or no interventions in NAFLD patients of all ages, considering baseline variations and associations between specific patient characteristics and vitamin E efficiency, along with the effect of the duration of the supplementation.

## 2. Materials and Methods

This systematic review has been registered in PROSPERO (registration number CRD42023392081) and the results are reported in accordance with the Preferred Reporting Items for Systematic Reviews and Meta-Analyses (PRISMA) statement as well as the standards of the Cochrane Handbook for Systematic Reviews of Interventions [21].

### 2.1. Literature Search Strategy

Three major databases, specifically MEDLINE, the Cochrane Central Register of Controlled Trials (CENTRAL), and Embase, were searched for randomized controlled trials testing the effect of vitamin E supplementation on ALT and AST when provided to NAFLD/NASH patients. The following keywords were used: “NASH”, “NAFLD”, “nonalcoholic steatohepatitis”, “nonalcoholic fatty liver disease”, “liver steatosis “, “fatty liver”, “vitamin E”, “alpha-tocopherol”, and “tocotrienol”. We also searched for unpublished trials and those in progress using the database of the National Institute of Health (ClinicalTrials.gov) (accessed on 25 April 2023). Authors of relevant trials whose results were not published were contacted via email. The bibliography of studies included was also searched in an attempt to identify possible additional studies that were not retrieved through the literature search. All manuscripts (original studies published in journals or as conference proceedings) published before 28 April 2023 were considered.

### 2.2. Eligibility Criteria and Study Selection

We aimed to find all randomized controlled trials testing vitamin E supplementation in NAFLD/NASH patients. Specifically, the inclusion criteria were: (1) patients diagnosed with NAFLD/NASH according to the international definitions [4,5], regardless of their age, gender, and ethnic origin; (2) the intervention included vitamin E (tocopherol, tocotrienol, or mixed forms), regardless of dosage, duration of supplementation, or route of administration. Co-interventions were considered eligible only if given equally in intervention and control arms; (3) patients in the control arm were under placebo, lifestyle modification advice, or no intervention; and (4) changes in serum ALT and AST were reported as mean differences (MD) [with or without their standard deviations (SD)]. Reasons for exclusion included co-morbidity with other liver diseases (including alcoholic steatohepatitis, viral or autoimmune hepatitis, drug-induced liver disease, cholestatic disorders, α-1-antitrypsin deficiency, and hemochromatosis) and intervention through changes in vitamin E intake from diet only (and not supplementation). Observational studies, as well as in vitro and animal studies or conference abstracts, were also excluded. No restrictions were imposed regarding the study’s language, geographical location or publication date. The titles/abstracts and full text of the studies identified in the original search were reviewed independently by two authors (SV and PO) in accordance with the exclusion criteria. Disagreements were resolved via consensus or after discussion involving a third author (AN).

### 2.3. Data Extraction and Assessment of the Risk of Bias

Information was extracted using a standardized data collection form and included: name of the first author, year of publication, identification number for each trial along with the corresponding website where the trial was registered (if available), country of origin, intervention and control groups, number of participants per group, age range of participants, dosage of vitamin E, frequency per day, co-interventions, study duration, and treatment outcome measures. Regarding the dosage of supplementation, vitamin E is currently listed on the new Nutrition Facts and Supplement Facts labels in mg [22] and the US Food and Drug Administration (FDA) has required manufacturers to use these new labels starting in January 2020. The conversion rule between mg and IU is: 1 mg of alpha-tocopherol is equivalent to 1.49 IU of the natural form or 2.22 IU of the synthetic form [22]. Authors of eligible studies whose results did not include adequate information for the calculation of the pooled estimates were contacted via email but no reply was provided. When there was not enough information available to calculate the standard deviations (SD) for the changes (MD) in enzymes levels, we followed the methodology for imputing SDs for changes from baseline suggested by Higgins et al. [21]. In particular, we computed the mean correlation coefficient (a number that describes how similar the baseline and final measurements were across participants) from the studies that had all the necessary information available and then we imputed the change-from-baseline SD in the other studies, making use of the imputed correlation coefficient. Sensitivity analyses were further carried out to assess the impact of the choice of different values of correlation, to determine whether the overall result of the analysis is robust to the use of imputed correlation coefficients.

The internal validity of eligible studies was assessed independently by two authors (SV and PO) using the Cochrane RoB2 assessment tool [21]. In case of disagreement, consensus was reached involving a third author (AN). The five domains through which bias might be introduced are: (1) bias arising from the randomization process; (2) bias due to deviations from intended interventions; (3) bias due to missing outcome data; (4) bias in measurement of the outcome; and (5) bias in selection of the reported result. Each domain could be characterized as at low risk of bias; causing some concerns; or at high risk of bias and each study could be judged to be at low risk of bias; at high risk of bias; or to raise some concerns [23].

### 2.4. Statistical Analysis

The analysis was conducted with the use of RevMan 5.4 [24]. A meta-analysis was performed to assess the effect of vitamin E supplementation on the MD of the ALT and the AST levels between the treatment and control groups. All data were continuous (presented as MD and SD) and were pooled as weighted MD with 95% confidence intervals (CI). The analysis was carried out for all eligible studies by the country of origin (Asian or non-Asian) followed by subgroup analyses including studies among children or adults as well as for the implication of lifestyle interventions or not and the duration of the studies. Additional sensitivity analyses were performed for each outcome by excluding each single study (leave-one-out method) in order to evaluate its influence on the pooled effect estimates. Heterogeneity was assessed with Cochran’s Q statistic and quantified using the I^2^ statistic, which indicated the proportion of variability across studies that was due to heterogeneity rather than sampling error. Values of 0–40%, 30–60%, 50–90%, and 75–100% represented low, moderate, substantial, and considerable heterogeneity; respectively [25]. Since participant characteristics and clinical settings differed greatly among studies, random effect models and the inverse variance method were applied in all cases. The possibility of publication bias was explored through the visualization of funnel plots and all results were regarded as statistically significant if *p* < 0.05.

## 3. Results

The study selection process is described in Figure 1. Our search retrieved 5686 unique citations from three electronic databases. After excluding duplicates and following title and abstract screening, we screened for eligibility for 65 full text articles. Of them, 53 articles were excluded after full text screening, and 12 reports of 12 RCTs (total *n* = 1229 patients) were reviewed and included in this meta-analysis (PRISMA flowchart; Figure 1).

The main characteristics of the eligible studies are presented in Table 1. All studies included were performed between 2006 and 2020; two trials were conducted in Europe [18,26], four trials in the USA [16,19,27,28] and six trials in Asia [29,30,31,32,33,34]. In seven studies [16,19,26,28,29,31] supplementation was combined with advice for lifestyle modification to both experimental and control groups (Table 1). With the exception of two studies [26,34], patients in control groups received a placebo similar to the supplement of the experimental group. One study [18] included a co-intervention to vitamin E, namely UDCA (ursodeoxyholic acid) that was given in equal doses to both groups. The follow-up period ranged from 4 to 96 weeks. Eight studies included adult populations and four included pediatric populations (Table 1). Two studies were conducted in mixed NAFLD and NASH pediatric populations [19,34].

### 3.1. Risk of Bias Assessment

A summary of the risk of bias assessment is presented in Appendix A. Using the Cochrane RoB2 tool, three [18,26,34] of the twelve trials were at high risk of bias because of insufficient description in multiple domains of the RoB2 tool [18] or concerns about the randomization processes (open-label process through distribution of participants to interventions) [26,34]. Five trials [29,30,31,32,33] were judged to be of some concern, and finally, four trials [16,19,27,28] were evaluated to be at low risk of bias.

### 3.2. Outcomes

#### 3.2.1. Effect of Vitamin E Supplementation on Aminotransferases

Overall, vitamin E supplementation had a significant effect on ALT levels [pooled MD = −13.06 IU/L favoring the experimental group, 95% CI (−19.17, −6.96), *I*^2^ = 94%] while it was also associated with a reduction in AST in patients in the experimental group [MD = −6.00 IU/L, (−9.51, −2.48), *I*^2^ = 68%] (Figure 2). In studies among Asian populations [29,30,31,32,33,34] vitamin E resulted in a decrease in ALT levels [MD = −14.35 IU/L, (−23.88, −4.82), *I*^2^ = 97%] and was also effective in AST reduction, inducing a mean change of 6.51 IU/L [MD = −6.51 IU/L, (−11.35, −1.67), *I*^2^ = 78%] (Figure 2). Vitamin E also had a beneficial impact on both ALT and AST levels in non-Asian populations [16,18,19,26,27,28]: regarding ALT, patients who received vitamin E had a greater mean reduction than controls [ALT MD = −9.57 IU/L, (−12.20, −6.95), *I*^2^ = 0%], whereas AST levels also reduced in the group of vitamin-E-receiving patients, although the finding was of lower precision [AST MD = −5.60 IU/L, (−11.48, 0.28), *I*^2^ = 63%] (Figure 2). The analysis was repeated after excluding the study of Wang et al. (2008) which was of particular concern regarding its high RoB score and since it appeared as an outlier. After the exclusion of this study, the conclusion for ALT remained unchanged but the MD was smaller [MD = −6.99 IU/L (−9.63, −4.35)]. The effect of vitamin E on AST remained but was attenuated [MD = −4.65 IU/L (−7.44, −1.86)]. A leave-one-out sensitivity analysis was carried out to examine the effect of each study on the synthesis outcome. With the exception of the Wang et al. (2008) study, no other trial was particularly influential and the inference for both transaminases was not affected. Funnel plots of the standard errors versus the MDs were examined for all outcomes; no publication bias was detected visually for ALT and AST (Appendix A).

#### 3.2.2. Subgroup Analysis

Apart from the ethnicity of the patients, subgroup analyses were performed according to their age, combined lifestyle interventions, as well as the duration of the therapy.

##### Effect of Vitamin E in Patients of Different Ages

In eight studies [16,18,27,28,29,30,32,33], patients were adults and vitamin E administration was shown to reduce both serum ALT [MD = −7.54 IU/L, 95% CI (−10.21, −4.87), *I*^2^ = 44%] and AST [MD = −5.58 IU/L, (−8.72, −2.43), *I*^2^ = 60%] (Figure 3). The pooled analysis of the four studies that included children and adolescents [19,26,31,34] indicated that vitamin E supplementation can cause a mean decrease in ALT equal to 22.71 IU/L favoring the experimental group [(−42.13, −3.29), *I*^2^ = 98%]. Among children receiving vitamin E, AST levels were reduced by 9.95 IU/L [(−26.99, 7.10), *I*^2^ = 81%] (Figure 3). During the leave-one-out sensitivity analysis, the pooled estimate did not change for either ALT nor AST with one exception: the exclusion of the study of Wang et al. (2008) led to a much smaller reduction of ALT among children [MD = −5.40 IU/L, (−12.47, 1.67)] and to a slight increase in AST levels [MD = 0.34 IU/L,(−4.66, 5.35)].

##### Effect of Vitamin E by Lifestyle Interventions

Among the included studies, seven [16,19,26,28,29,31,33] provided the same advice for lifestyle modification to their experimental and control groups (Table 1). Based on studies that combined lifestyle advice, vitamin E was associated with a mean decrease in ALT equal to 7.35 IU/L [95% CI (−11.27, −3.43), *I*^2^ = 78%] and a mean decrease in AST equal to 4.00 IU/L [(−7.05, −0.94), *I*^2^ = 43%] (Figure 4). Regarding changes in ALT, the pooled estimate of the five studies that did not include lifestyle interventions was MD = −19.36 IU/L [(−37.30, −1.41), *I*^2^ = 97%] (Figure 4), with AST levels decreasing similarly among participants receiving only vitamin E supplements as compared to their controls [MD = −12.49 IU/L, (−22.30, −2.68), *I*^2^ = 83%] (Figure 4). During the sensitivity analysis, the mean decrease for ALT among the studies that did not include lifestyle interventions loses its significance when one of the studies of Basu et al. (2014) [27], Ekhlasi et al. (2016) [30], or Magosso et al. (2013) [32] is excluded, while the exclusion of the study of Wang et al. (2008) [34] leads to a significant but smaller mean decrease in ALT [MD = −7.11 IU/L (−9.89, −4.33)]. After the exclusion of the study of Wang et al. (2008) [34], in which no lifestyle advice was given to the patients, the pooled reduction in AST levels was smaller [MD = −6.66 IU/L (−13.93, 0.61)].

##### Effect of Vitamin E according to the Duration of the Intervention

We also examined the effect of vitamin E supplementation according to the duration of the intervention, using 24 weeks (median duration) as a cut-off. Seven trials had a duration range from 4 to 24 weeks and five trials from 48 to 96 weeks [16,18,19,28,32] (Table 1). In studies lasting up to 24 weeks [26,27,29,30,31,33,34], vitamin E supplementation resulted in a decrease in ALT levels and the magnitude was similar to that of the overall polled effect [MD = −13.77 IU/L, 95% CI (−21.70, −5.83), *I*^2^ = 97%]. The effect on AST levels was, however, smaller in studies of shorter duration [MD = −6.25 IU/L, (−11.38, −1.11), *I*^2^ = 80%] (Figure 5). Vitamin E had a beneficial impact on both ALT and AST levels in studies lasting between 48 and 96 weeks [ALT MD = −10.75 IU/L, (−17.15, −4.36), *I*^2^ = 32% and AST MD = −5.83 IU/L, (−10.57, −1.09), *I*^2^ = 34%] (Figure 5). The exclusion of the study of Wang et al. (2008) [34] that only lasted for 4 weeks resulted in smaller reductions in both ALT levels [MD = −6.10 IU/L (−9.12, −3.09)] and AST levels [MD = −4.02 IU/L (−7.80, −0.23)].

## 4. Discussion

A meta-analysis of randomized and controlled trials among NAFLD adult patients testing the impact of vitamin E supplementation alone on the serum levels of aminotransferases AST and ALT showed that vitamin E supplementation ranging from 400 IU (268 mg) to 800 IU (536 mg) consistently reduced serum ALT and AST levels compared with a placebo, with the effect being more prominent in the ALT levels. The beneficial effect of vitamin E seemed to be independent of ethnicity, the concurrent interventions to improve overall dietary intake, and increase physical activity. Furthermore, the beneficial effect of vitamin E supplementation on ALT and AST did not differ between studies lasting for 48 weeks or more as compared to those lasting for up to 24 weeks.

Although distinct pathways of NAFLD pathophysiology among Asian races cannot be excluded [35,36], our results do not support the differential effects of vitamin E supplementation according to the patients’ ethnicity. Interestingly, the age-specific analysis revealed that children and adolescents with NAFLD could benefit more from increased vitamin E intake regarding ALT but not AST levels. One eligible study (Wang et al., 2008) [34] carried out among adolescents in China tested a small dosage (100 mg of vitamin E per day) and only lasted for four weeks. The study was judged to be at a high risk of bias due to deviations in the randomization process, and reported a substantially greater effect particularly on ALT levels. Excluding this study from the analysis lead to changes in ALT and AST levels of closer magnitude. The elevated baseline aminotransferases of its participants, indicative of NASH, make the evaluation of ALT and AST as treatment outcomes complicated, and partially explains why the MD of ALT is an outlier compared with the results of the rest of the included studies.

The meta-analysis of Amanullah et al. (2019) [37] included both adults and children and did not report a significant decrease in aminotransferases levels in the pediatric age group. Of note, although this review included patients of all ages, the meta-analysis was performed only in studies including children and one trial was included twice. In another meta-analysis [38] of 15 RCTs with NAFLD patients of all ages, the authors considered short-, intermediate-, and long-term follow-ups, and concluded that the group receiving vitamin E had a significant decrease in ALT and AST levels among adults. Interestingly, among children the significant change in biochemical parameters appeared at longer follow-up (i.e., after 12 months). The favorable effect of vitamin E on children, also observed in our analysis, could be partially explained by the important role of oxidative stress in pediatric NAFLD. Studies have reported increased levels of circulating biomarkers of oxidative stress [39,40]. Up to 83% of children with NAFLD show signs of oxidative injury, as evaluated via elevated protein carbonyls and an increased hepatocyte nuclear staining for 8-hydroxy-2-deoxyguanosine (8-OHG), a lipid peroxidation end-product and marker of oxidative DNA damage [40].

In our analyses, the effect on AST seems to be slightly weaker overall as well as in subgroup analyses. Our results are in accordance to those of the two recent meta-analyses of vitamin E’s effect on NAFLD [20,38] with ALT levels being further decreased compared with AST levels in all subgroups. However, our findings rely on studies that allow for the estimation of the independent vitamin E effect. The meta-analysis of Abdel-Maboud et al. (2020) [38], for instance, includes trials that tested mixed supplements, raising questions on whether the benefit can be attributed to vitamin E or to a synergistic effect of all the supplements tested.

After the exclusion of the Wang et al. (2008) [34] study of a very short duration, the effect of vitamin E on aminotransferases was greater in interventions lasting between 48 and 96 weeks. This result is in line with the conclusions of the meta-analysis by Abdel-Maboud et al. (2020) [38], who reported that the overall effect on ALT and AST in adults was different after 6, 12, 18, and 24 months of follow-up, showing a stronger beneficial effect of vitamin E with longer follow-up periods.

The dose of vitamin E varied widely among trials and two scientific bodies suggest 800 IU as a daily dose of vitamin E for NAFLD patients [4,5], which is much higher than the daily recommended dose of 20–30 IU for healthy adults [22]. High doses of vitamin E, however, may still provoke some concern. In an attempt to examine its role in tumorigenesis, two large trials (HOPE-TOO Trial and Women’s Health Study) failed to show a benefit from vitamin E supplementation in any tumor incidence [41,42]. A large RCT, the SELECT trial, using vitamin E with or without selenium was initiated in 2001 and discontinued in 2008 after evidence showing that vitamin E supplementation increased prostate cancer risk by up to 17% [43]. In addition, an extensive meta-analysis of 57 RCTs examining the effect of vitamin E on all-cause mortality concluded that vitamin E doses of up to 5500 IU/day were not associated with a lower risk of premature death [44].

Regarding the form of vitamin E, in three of the included studies investigators used α-tocopherol, one δ-tocotrienol, and one mixed tocotrienols, while in seven studies the exact form was not mentioned. As a result, we could not examine the role of different forms of vitamin E in NAFLD and this is a limitation of our study. A previous meta-analysis [20] indicated that α-tocopherol was associated with a decrease in the aminotransferases while tocotrienol only significantly decreased ALT and not AST. Other limitations of our study are the relatively small sample size of the included RCTs together with the differences in dose and duration of supplementation that might have affected the accuracy of the results. Although aminotransferases are quite frequently used as non-invasive serological markers of NAFLD activity, they remain both insensitive and non-specific for NAFLD [45]. However, patients with NASH and advanced fibrosis have frequently abnormally elevated ALT [46], and ALT serum levels have been found to be positively correlated with NAFLD activity score (NAS), liver triglyceride content, Homeostatic Model Assessment for Insulin Resistance (HOMA-IR), and fasting insulin, making it a good predictor of NAFLD [47]. Along with BMI, waist circumference, and serum triglyceride, aminotransferases have also been used in NAFLD risk scoring models [48], giving us the rationale behind our choice as outcomes of interest.

Our review features several strengths. We have included trials that allow for the evaluation of vitamin E supplementation on liver enzymes and we attempted to detect possible sources of bias in all included studies, making it the most up-to-date systematic review and meta-analysis of the topic. We have further undertaken a number of subgroup analyses trying to identify factors that may contribute to the heterogeneity observed in the overall analyses. The mixed choices of supplementation, ages, and study origins were common in previous meta-analyses on the topic [38,49,50]. As a result, this is the first study to examine the effect of vitamin E versus placebo in patients with NAFLD including the most recent clinical trials, further accompanied by extensive subgroup analyses concerning the age group, the study origin, the duration of the intervention, and the concomitant provision or lifestyle advices.

## 5. Conclusions

In conclusion, we found evidence that vitamin E supplementation can contribute to a decrease in serum ALT and AST levels among NAFLD adult patients. Further clinical trials of higher internal validity investigating the optimal dose, the synthetic form of the vitamin, and the duration of therapy together with recording potential adverse effects are needed.

## Figures and Tables

**Figure 1 nutrients-15-03733-f001:**
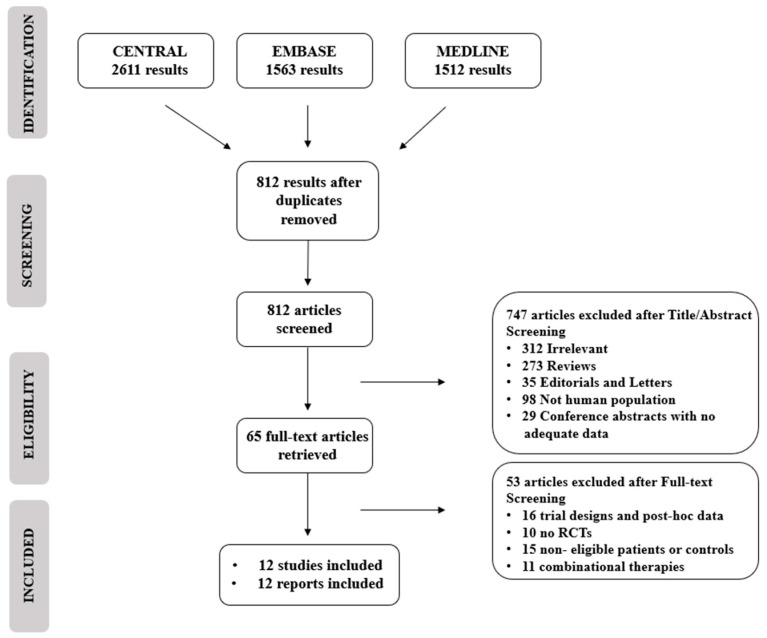
PRISMA flowchart and reasons for exclusion of studies.

**Figure 2 nutrients-15-03733-f002:**
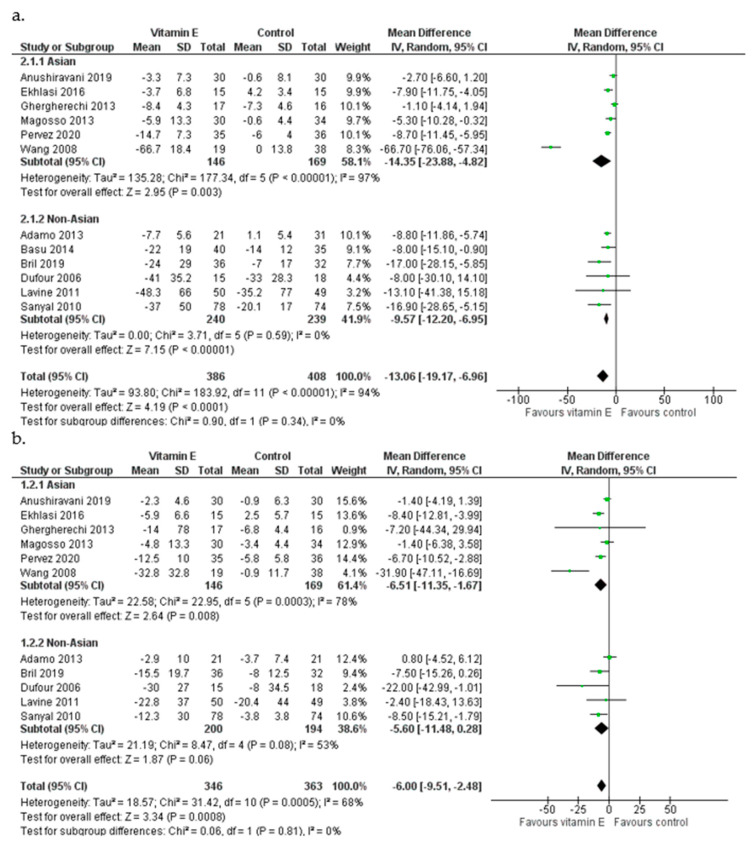
Meta-analysis of randomized and controlled trials assessing the effect of vitamin E supplementation on ALT (**a**) and AST (**b**) levels in Asian and non-Asian patients with NAFLD [16,18,19,26,27,28,29,30,31,32,33,34].

**Figure 3 nutrients-15-03733-f003:**
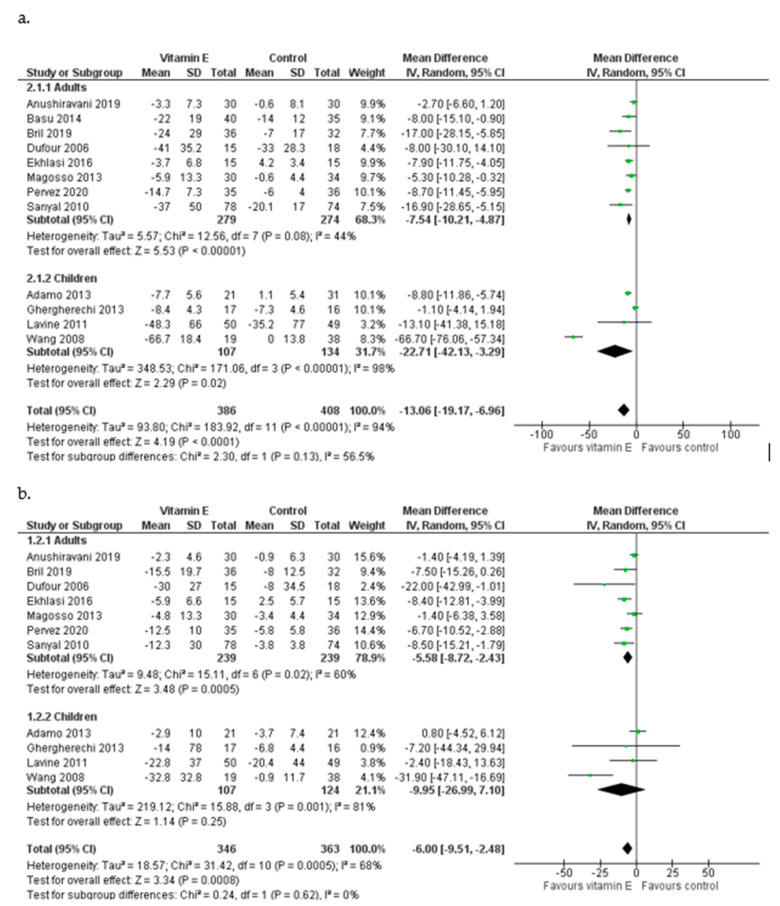
Meta-analysis of randomized and controlled trials assessing the effect of vitamin E supplementation on ALT (**a**) and AST (**b**) levels in adults and children with NAFLD [16,18,19,26,27,28,29,30,31,32,33,34].

**Figure 4 nutrients-15-03733-f004:**
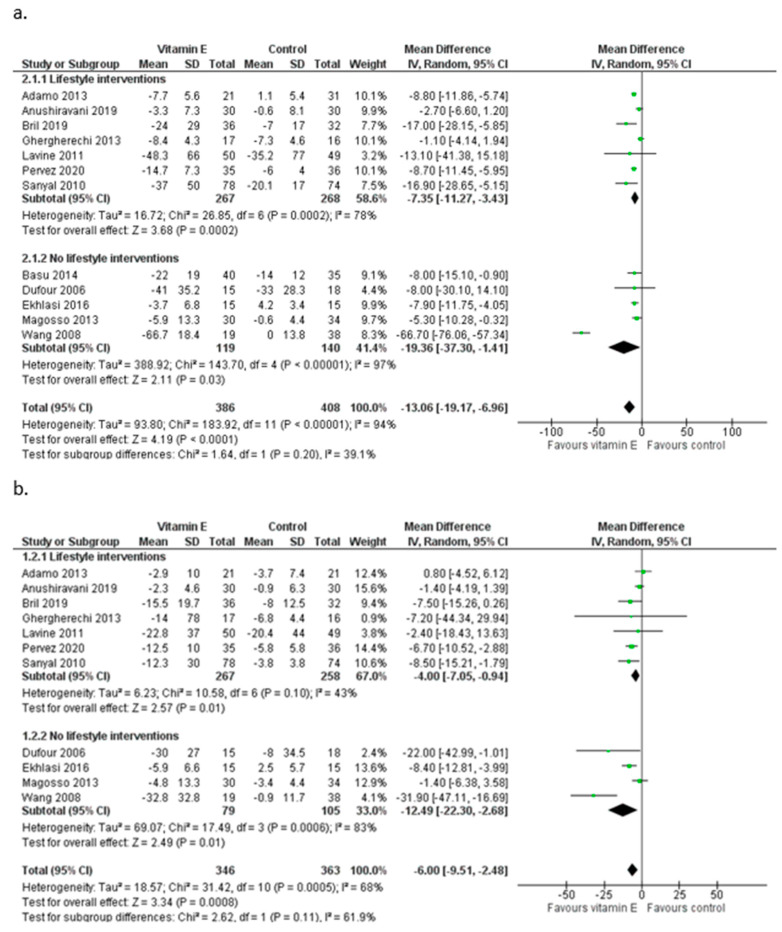
Meta-analysis of randomized and controlled trials assessing the effect of vitamin E supplementation on ALT (**a**) and AST (**b**) levels with or without additional lifestyle interventions in patients with NAFLD [16,18,19,26,27,28,29,30,31,32,33,34].

**Figure 5 nutrients-15-03733-f005:**
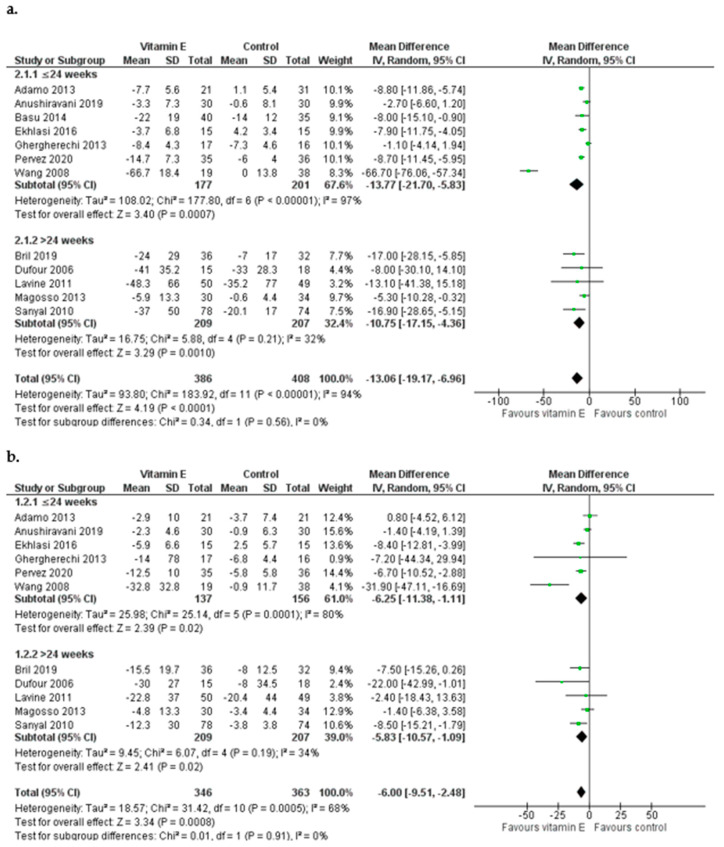
Meta-analysis of randomized and controlled trials assessing the effect of vitamin E supplementation on ALT (**a**) and AST (**b**) levels by duration of the intervention in patients with NAFLD [16,18,19,26,27,28,29,30,31,32,33,34].

**Table 1 nutrients-15-03733-t001:** Characteristics of randomized controlled trials assessing the effect of vitamin E supplementation on outcome related to NAFLD/NASH.

Author, Year (Clinical Trial Registration Number)	Country	Population	Number of Study Participants (Number in Each Group, I: Intervention, C: Control)	Intervention	Daily Dosage ^†^	Study Duration (Weeks)	Comparison
Adamo, 2013 [26]	Italy	Children	42 (I:21, C:21)	vitamin E (D-α-tocopherol) + lifestyle interventions ^1^	600 mg	24	lifestyle interventions ^1^
Anushiravani, 2019 [29] IRCT201705016312N4 (http://www.irct.ir/) (accessed on 23 April 2023)	Iran	Adults	150 (I:30, C:30)	vitamin E + lifestyle interventions ^2^	400 IU	12	placebo + lifestyle interventions ^2^
Basu, 2014 [27]	USA	Adults	155 (I:40, C:35)	vitamin E	700 IU	24	placebo
Bril, 2019 [28] NCT01002547 (http://www.clinicaltrials.gov/) (accessed on 25 April 2023)	USA	Adults	105 (I:36, C:32)	vitamin E + lifestyle interventions ^2^	400 IU	72	placebo + lifestyle interventions ^2^
Dufour, 2006 [18]	Switzerland	Adults	48 (I:15, C:18)	vitamin E + UDCA *	800 IU	96	placebo + UDCA
Ekhlasi, 2016 [30] IRCT201111082709N22 (http://www.irct.ir/) (accessed on 23 April 2023)	Iran	Adults	60 (I:15, C:15)	vitamin E (α-tocopherol)	400 IU	8	placebo
Ghergherehchi, 2013 [31]	Iran	Children 4–15 yo	33 (I:17, C:16)	vitamin E + lifestyle interventions ^3^	400 IU	24	placebo+ lifestyle interventions ^3^
Lavine, 2011 [19] NCT00063635 (http://www.clinicaltrials.gov/) (accessed on 25 April 2023)	USA	Children 8–17 yo	173 (I:50, C:49)	vitamin E (α-tocopherol) + lifestyle interventions	800 IU	96	placebo + lifestyle interventions ^4^
Magosso, 2013 [32] NCT00753532 (http://www.clinicaltrials.gov/) (accessed on 25 April 2023)	Malaysia	Adults	87 (I:30, C:34)	vitamin E (mixed tocotrienols)	400 mg	48	placebo
Pervez, 2020 [33] SLCTR/2015/023 (https://slctr.lk/ SLCTR/2015/023) (accessed on 23 April 2023)	Pakistan	Adults	71 (I:35, C:36)	vitamin E (δ-tocotrienol) + lifestyle interventions ^5^	600 mg	24	placebo + lifestyle interventions ^5^
Sanyal, 2010 [16] NCT00063622 (http://www.clinicaltrials.gov/) (accessed on 25 April 2023)	USA	Adults	247 (I:78, C:74)	vitamin E + lifestyle interventions ^6^	800 IU	96	placebo + lifestyle interventions ^6^
Wang, 2008 [34]	China	Children aged 10–17 yo	76 (I:19, C:38)	vitamin E	100 mg	4	no intervention

^†^ 1IU of vitamin E is equivalent to 0.67 mg of alpha tocopherol, * UDCA: ursodeoxyholic acid, ^1^ nutritional and exercise recommendations, ^2^ 500 kcal/day deficit from weight-maintaining caloric intake, ^3^ low-calorie diet from 1300 to 1800 Kcal based on the individual requirements and increased physical activity, including aerobic exercise and walking for 2 h every day, ^4^ patients and parents were provided standard-of-care advice on diet and exercise at each visit by physicians and dieticians, ^5^ advice for a low-fat diet and regular physical activity, ^6^ a standardized set of pragmatic recommendations about lifestyle changes and diet.

## Data Availability

The data presented in this study are available on request from the corresponding author.

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
