# Peer review of "The Effect of Vitamin E Supplementation on Serum Aminotransferases in Non-Alcoholic Fatty Liver Disease (NAFLD): A Systematic Review and Meta-Analysis"

_nutrients, 2023, doi:10.3390/nu15173733_

Round 1

Reviewer 1 Report

This study is a comprehensive review that consolidates the findings on the efficacy of vitamin E in treating NASH, particularly in ameliorating serum ALT elevation among patients. Overall, the manuscript is well-crafted, and the statistical methods have been appropriately chosen for the investigation. However, there are a few minor issues that require the attention:

  1. In Table 1, if available, please ensure that you include the identification number for each trial along with the corresponding website where these trials were registered.
  2. Within the same table, there seems to be an inconsistency in the total number of participants in the study conducted by Adamo et al. While you've listed the total as 24 participants, the breakdown indicates 21 participants in the intervention group and 21 participants in the control group. Kindly rectify this error.
  3. An observation has been made regarding Figure 4. The values presented in panel a (ALT) and panel b (AST) are identical. This suggests that there might have been errors during the manuscript's preparation. Please review and correct these discrepancies.
  4. To maintain consistency, kindly replace all instances of "[]" with "()" when referring to confidence intervals (CI) throughout the manuscript.
  5. On page 205, in line 206, it is recommended to replace "I2" with "I²" to correctly represent the "I-squared" statistic.

Addressing these points will enhance the accuracy and clarity of your study, ensuring that the findings are presented in a precise and reliable manner.

Author Response

Reviewer #1

  1. This study is a comprehensive review that consolidates the findings on the efficacy of vitamin E in treating NASH, particularly in ameliorating serum ALT elevation among patients. Overall, the manuscript is well-crafted, and the statistical methods have been appropriately chosen for the investigation.

We thank the Reviewer for the kind words. No revisions required in this paragraph.

  1. In Table 1, if available, please ensure that you include the identification number for each trial along with the corresponding website where these trials were registered.

The available identification numbers and the corresponding websites where trials have been registered are now included. We thank the Reviewer for pointing this out. (REVISED: Table 1)

  1. Within the same table, there seems to be an inconsistency in the total number of participants in the study conducted by Adamo et al. While you've listed the total as 24 participants, the breakdown indicates 21 participants in the intervention group and 21 participants in the control group. Kindly rectify this error.

We are grateful to the Reviewer for the astute reading. The entry has now been revised (REVISED: Table 1)

  1. An observation has been made regarding Figure 4. The values presented in panel a (ALT) and panel b (AST) are identical. This suggests that there might have been errors during the manuscript's preparation. Please review and correct these discrepancies.

We are again thankful to the Reviewer for observing this oversight, which has now been corrected. (REVISED: Figure 4, panel b)

To maintain consistency, kindly replace all instances of "[]" with "()" when referring to confidence intervals (CI) throughout the manuscript.

On page 205, in line 206, it is recommended to replace "I2" with "I²" to correctly represent the "I-squared" statistic

Both typos indicated by the Reviewer have been amended throughout the manuscript. REVISED

Reviewer 2 Report

This study lacks novelty. The subject has been extensively described and a large number of articles of similar type have been published. Therefore, we feel that this contribution cannot add significantly to our understanding. In addition, many published meta-analyses have already illustrated the role of vitamin E more fully through network meta-analysis, and therefore exploring only the effect on ALT/AST is not sufficiently comprehensive.

Meta-analysis of the effects of vitamin E on NAFLD published after 2020:(PMID: 35280867, PMID: 36183848, PMID: 34280304, PMID: 36686141, PMID: 33272889, PMID: 34435378, PMID: 34435378, PMID: 34435378, PMID: 34435378,  PMID: 36689199, PMID: 36689199, PMID: 35970684)

In addition, vitamin E has been recognized as having an ameliorative effect on NAFLD/NASH and may be considered a first-line, off-label treatment for which homogenous studies may no longer be necessary. Research should focus on emerging areas of improvement with these drug therapies.

Author Response

The Reviewer is right in pointing out that vitamin E supplementation in combination with therapy has an ameliorative effect in NAFLD patients and four out of the eight meta-analyses proposed have addressed this objective, i.e. to assess the effect of therapy with or without vitamin E supplementation in treating patients with NAFLD. Of note that the other four meta-analyses mentioned by the Reviewer have examined the effect of vitamin E as a single agent on NAFLD-related outcomes, but studies have included a selection of large RCTs [namely Sanyal et al (2010), Lavine et al (2011), Bril et al (2019)] without explaining however why other RCTs that appear to have met their inclusion criteria were not considered.

Hence, we would respectfully like to point out to the Reviewer that the aim of our meta-analysis is different as we have opted to analyse the results of randomised trials in which vitamin E supplementation was the only intervention in treating NAFLD and patients received no other treatment. Moreover, we have undertaken a comprehensive analysis including all eligible trials and performed sensitivity and sub-group analyses to acquire a concrete understanding of the findings. We have now inserted sentences in the text to have our objective clearly articulated. (REVISED: Introduction line 76, Discussion, line 325)

Reviewer 3 Report

This is a reasonably good update of the subject, and the conclusions match the data, such as they are.  However, there should be recognition that the studied themselves were not all that reliable; so may with > 50% drop-outs (according to Table 1).  Even the best analysis of unreliable data produces an unreliable conclusion.  That said, the authors' conclusion is kept from the reader until the nearly last sentence of the manuscript; the Abstract leaves the reader hanging.

Author Response

We are grateful to the Reviewer for the kind words and the thoughtful comment. Following this suggestion, we have revised the abstract and the conclusion of our study accordingly. (REVISED: Abstract lines 16-17, and Discussion, line 418)

Round 2

Reviewer 2 Report

Given the attitude and explanation of the response, it could be considered acceptable.